# Screen Time and Its Association with Vegetables, Fruits, Snacks and Sugary Sweetened Beverages Intake among Chinese Preschool Children in Changsha, Hunan Province: A Cross-Sectional Study

**DOI:** 10.3390/nu14194086

**Published:** 2022-10-01

**Authors:** Jiaqi Huo, Xiaoni Kuang, Yue Xi, Caihong Xiang, Cuiting Yong, Jiajing Liang, Hanshuang Zou, Qian Lin

**Affiliations:** 1Department of Nutrition Science and Food Hygiene, Xiangya School of Public Health, Central South University, 110 Xiangya Road, Changsha 410078, China; 2Department of Child Care, Changsha Maternal and Child Health Care Hospital, 416 Chengnan East Road, Yuhua District, Changsha 410007, China; 3Department of Epidemiology, School of Public Health, Sun Yat-sen University, Guangzhou 510275, China

**Keywords:** preschooler, screen time, vegetables and fruits, snacks, sugary sweetened beverages (SSBs)

## Abstract

(1) Introduction: Screen time may influence preschoolers’ food consumption. However, there is limited evidence regarding preschoolers, especially in China. The aim of this cross-sectional study was to investigate the association between screen time and the consumption of vegetables, fruits, snacks, and sugar sweetened beverages (SSBs). (2) Methods: Participants (1567 caregivers) were recruited from six kindergartens in Hunan, China. Caregivers completed the questionnaire, which included the food frequency questionnaire (FFQ) and questions regarding their children’s daily screen time. (3) Results: The mean screen time of preschoolers was 1.36 ± 1.26 h, and the proportion of children who spent more than one hour on screens was 54.3% in the overall sample. Children with longer screen time consumed vegetables and fruits less frequently, while having a higher consumption of snacks and SSBs. After adjustment of sociodemographic confounders, children’s eating behaviors and parental feeding practices, the association of screen time with vegetables and SSBs still remained significant. (4) Conclusions: Screen time exposure needs to be monitored in preschool children, which was negatively associated with their consumption of vegetables and fruits, whereas it was positively associated with snacks and SSBs. Future research should focus more on the impact of screen time on children’s unhealthy behaviors and dietary patterns.

## 1. Introduction

China is undergoing a period of rapid nutritional transition, with an improvement in diet quality, along with an increasing consumption of processed foods rich in sugar and fat. Excessive intake of snacks and sugar sweetened beverages (SSBs) along with insufficient intake of vegetables and fruits (F&V) are common among preschool children in China [1]. A study [2] showed that the median vegetable intake of preschoolers aged 3–5 years was 130.5 g/day, and the median fruit intake was 175.0 g/day; only 22.1% and 68.8% of children reached the standard of F&V intake. Other studies have also shown these problems, mainly the deficiency of F&Vs [3,4,5]. According to the dietary guidelines for Chinese residents (2022), preschoolers (3~6 years old) are recommended to eat vegetables and fruits every day. It is recommended to consume 150 to 300 g of vegetables each day, and 150 to 250 g of fruits every day. The guarantee of preschooler’s health is adequate intake of F&V, whereas consuming too much energy-dense food (snacks and sugar sweetened beverages) will negatively affect their diet quality, leading to overweight and obesity, and hence increasing the risk of chronic diseases in adulthood [6,7,8]. Thus, it is of great importance to boost dietary quality to improve the F&V intake and decrease snacks and SSB consumption. In addition to this, studies have shown that there is a strong connection between screen time and dietary quality (F&V intake, snacks and SSB consumption) [6,7].

Screen time is an emerging factor for dietary behaviors among children. It refers to the amount of time spent on screens, such as computers, television, video games, smartphones, etc. [9]. Excessive screen time has been linked to increased body weight [10,11], shorter sleep duration [12], and mental health problems [11,12,13,14] in children. According to the World Health Organization’s recommendations, preschoolers should spend less than one hour per day on electronic screens [15]. A researcher in Canada concluded that the average screen time for 3-year-old children was 3.57 h per day, and for 5-year-olds, it was 1.55 h per day [16]. Another study conducted in Guangzhou found that 51.8% of Chinese preschoolers spent more than one hour on electronic screens per day [17]. Therefore, whether in China or elsewhere, preschoolers’ screen time exposure is of particular concern. It is crucial to monitor the current situation of preschoolers’ screen time exposure.

Many different studies have suggested that prolonged screen time exposure leads to decreased intake of fruits and vegetables, and increased consumption of snacks and sugary sweetened beverages (SSBs) [6,7,8,18,19]. A study conducted in Canadian children also revealed that television viewing was positively associated with energy-dense foods and drinks, as well as fast food consumption, and negatively associated with the consumption of fruits and vegetables [20], consistent with other studies [21]. However, most of these researches focused on school-aged children, adolescents [22], or were targeted to explore the hazards of excessive screen time [23,24]. Only a few studies had attempted to explore the association between preschoolers’ screen time and food consumption. that the pre-school period is an important early life stage for establishing good eating behaviors and habits that are crucial for adulthood health [25]. Thus, it is necessary to understand the current status of screen time in pre-school children and its relationship with food consumption.

To date, studies that concern the association between screen time and food consumption are limited in China. In addition, due to the implementation of the double reduction policy, the three-child policy and the epidemic of the COVID-19 virus, preschool children have more and more disposable time, and the problem of screen time exposure is worth paying attention to. Therefore, a cross-sectional study was designed to investigate the current situation of screen time in Changsha, Hunan Province, and explore the association between screen time and food consumption, to attract attention to the use of electronic screens and prevent the formation of unhealthy behaviors in preschool children.

## 2. Materials and Methods

### 2.1. Ethic Approvals

This study was approved by the Ethics Review Committee of the Xiangya School of Public Health, Central South University (No. XYGW 2020-105).

### 2.2. Study Design and Participants

This was a cross-sectional study. We adopted the method of random cluster sampling to select 6 kindergartens from all public kindergartens in Changsha City, Hunan Province, from January to March 2021. The inclusion criteria for parents and children were as follows: the participating parents (1) were responsible for their children’s diet for more than six months, (2) could read and fully understand the questionnaire, and (3) their children were between the ages of 2 and 7. Eligible children–parent pairs were enrolled in the study if the parents provided informed consent. Based on the study results, the proportion of preschooler children with vegetable intake < 6 times/week was 32.3% [26]. The sample size was calculated for 90% power and a two-sided 95% confidence interval, resulting in approximately 443 children–parent pairs when the dropout rate was 20%. A total of 1616 children–parent pairs took part in this study, and 1574 were eligible for the study after excluding those who failed to fill out the questionnaire. The response rate was 97.0%. Considering that the dietary guidelines for Chinese preschool children targets children aged 3 to 6 years, children younger than 2 years and older than 7 years were excluded, and the final number of children–parent pairs was 1567.

### 2.3. Data Collection

Data were collected through the Questionnaire Star, an online survey tool used to develop electronic questionnaires (www.wjx.cn accessed on 4 March 2021). Each questionnaire had a unique QR code. The teacher would issue the QR code of the electronic questionnaire for the parents to fill in. Information on the children’s screen time, eating behaviors, food neophobia, weekly food consumption frequencies, and feeding practices was collected, along with the characteristic of the children and their caregivers.

#### 2.3.1. Demographics and Anthropometrics

We collected general demographic information, including gender, date of birth, nationality, place of residence, left-behind status, number of children in the household, ethnicity and education level of primary caregivers, through a self-administered questionnaire. 

We collected data on parents’ self-reported height and weight of their children and performed BMI-for-age Z-score estimates. According to the World Health Organization reference standard for child growth, the weight status of children was classified as obese (BMI ≥ +2 SD), overweight (BMI ≥ +1 SD), normal weight (BMI > −1 SD to < +1 SD), and underweight (BMI ≤ −1 SD).

#### 2.3.2. Children’s Screen Time

Children’s screen time was collected through a self-made questionnaire that consisted of the following items:

(1). During the past few weekends, how much time did your child spend watching TV, e-reading, using mobile phones, computers, etc.?

(2). During the past few weekdays (Monday to Friday), how much time did your child spend watching TV, e-reading, using mobile phone, computer, etc.?

The weekday screen time and weekend screen time were transformed to average hours per day as follows: ((weekday hours * 5) + (weekend hours * 2))/7. Children should spend no more than 1 h on electronic screens every day, according to the recommendations of the exercise guide for preschool children in China [27]. Therefore, the average screen time were divided into <1 h or ≥1 h.

#### 2.3.3. Children Food Consumption Frequencies

The children’s food consumption frequencies for the past three months were collected using a 27-item semi-quantitative food frequency questionnaire (FFQ) [28,29], whose response frequencies were as follows: never or <once/month, 1~3 times/month, 1~2 times/week, 3~4 times/week, 5~6 times/week, once/day, twice/day, 3 times/day or more, and then converted to 0, 0.25, 0.5, 1.5, 3.5, 5.5, 7, 14, and 21 times per week, respectively. Considering the prevalence of marginal vitamin A deficiency was 31.53% in Chinese preschool children [30], we categorize vegetables and fruits into vitamin A-rich vegetables/other vegetables and vitamin A-rich fruits/other fruits to understand the current status of vitamin A-rich food intake among preschool children. According to the dietary guidelines for Chinese residents, children’s vegetable and fruit consumption was divided into ≥once/day and <once/day. The snack group included sweets, ice cream, honey, spicy food, fast food (KFC (Kentucky Fried Chicken, North Corbin, KY, USA), McDonald’s (Mc Donald’s Corporation, Chicago, IL, USA), fried chicken, etc.) and puffed fried food (potato chips, French fries, instant noodles, etc.). The sugary sweetened beverages (SSBs) group included fresh fruit and vegetable juice, and sugary drinks (carbonated drinks, milk tea, tea drinks, milk drinks, etc.). According to the snacks consumption guideline for Chinese children, snacks and SSB consumption was divided into ≥once/week and <once/week [31].

#### 2.3.4. Covariates

The covariates included parental feeding practices, children eating behaviors and food neophobia. Parental feeding practices were evaluated using the Chinese version of the Child Feeding Questionnaire (CFQ), which has been adapted and validated (Cronbach’s α coefficient = 0.705) [32,33]. The Chinese CFQ consisted of 26 items and 8 facets. The responses to each question are based on a 5-point Likert scale, expressing agreement (1 = disagree to 5 = agree), frequency (1 = never to 5 = always), concern (1 = unconcern to 5 = very concern), or weight evaluation (1 = markedly underweight to 5 = markedly overweight). The score for each subscale is the average score of the item to which it belongs. The higher the score of each subscale is, the more controlling the parents are in this facet. According to the mean score of each subscale, caregivers were grouped by median values as high or low levels based on their feeding practices.

The children’s eating behaviors were measured by a validated Chinese version of the Children’s Eating Behaviors Questionnaires (CEBQ, Cronbach’s α coefficient = 0.741) [34], which consists of 35 items and 8 facets. The responses of the scale include the following five options: never, rarely, sometimes, often and always, which are given 1~5 points, respectively. Except for food fussiness, the other children’s eating behaviors were divided into high or low level according to the median of the mean of each subscale. As for food fussiness, they were divided into high, medium and low according to the values of 3 and 3.33 [35].

Children’s food neophobia was measured by the validated Chinese version of the Child Food Neophobia Scale (CFNS, Cronbach’s α coefficient = 0.910) [36,37], which includes 6 items, with responses ranging from 1(“strongly disagree”) to 7 (“strongly agree”), and the scale score ranging from 6 to 48 points. The higher the score is, the more serious the children’s food neophobia behavior is. According to the median score of the scale, the children’s food neophobia level was divided into two groups, high and low.

### 2.4. Data Analysis

Data were analyzed using IBM SPSS 26.0 software (IBM Corp., Armonk, NY, USA) and R Studio (R Foundation for Statistical Computing, Vienna, Austria), and the figure was performed by GraphPad Prism 8.3.0 software (LA Jolla, CA, USA). The chi-squared and one-way AVOVA test were employed to analyze general demographic data for different categories or continuous variables. Binary logistic regression analyses were used to identify the association between screen time and vegetable, fruit, snack, and SSB consumption. The explanatory variables with *p* < 0.10 were included in the multivariate analysis using the backward stepwise method for binary logistic regression. We considered a two-tailed *p* < 0.05 statistically significant in all analyses.

## 3. Results

### 3.1. Screen Time

In this study, the mean screen time of preschoolers was 1.36 ± 1.26 h, with the median of 1 (0.5~1.79) hour per day. The proportion of children who spent more than one hour on electronic screens was 54.3% (Table 1). Among them, 836 were boys, accounting for 53.4%. The proportion of children aged 4 to 5 was 27.5%. The proportions of screen time among the age and the gender of preschoolers are shown in Figure 1a,b. There were no statistically significant differences between screen time, age and gender among preschool children (53.8% vs. 27.5% vs. 18.7%, *p* = 0.776; 53.4% vs. 46.6%, *p* = 0.800). Most of the children lived in cities (93.9%) and were not the only child in the family (61.4%). According to the self-reported height and weight of the children, 56.5% of the children were of normal weight, while the proportion of overweight and obese children was 22.0%. Most primary caregivers were mothers (88.8%), most of whom acquired college education or above (86.0%). Children’s screen time varied among primary caregivers at different educational levels (90.2% vs. 82.4%, *p* < 0.001).

### 3.2. Food Consumption Frequency of the Participants

For children, the consumption frequency of vegetable, fruits, snacks, and SSBs was 9.27 ± 7.66 times/week, 8.34 ± 6.93 times/week, 4.66 ± 6.40 times/week, and 1.21 ± 2.58 times/week, respectively. There were no statistically significant differences between food consumption and children’s age and gender (Table 2). The consumption frequencies of food intake varied at different levels of screen time (Figure 2). The lower the frequency of vegetable consumption, the more time they spent on screens (*p* = 0.002). Moreover, frequent snacks (*p* = 0.001) and SSB (*p* = 0.001) consumption was correlated with longer screen time.

### 3.3. Associations between Screen Time and Consumption of Vegetables, Fruits, Snacks and Sugary Sweetened Beverages

The children with screen time ≥ 1 h were more likely to have low consumption of vegetables (crude OR = 1.371, 95% CI: 1.112~1.690, *p* < 0.01) and lower fruit intake (crude OR = 1.233, 95% CI: 1.008~1.509, *p* < 0.05). However, they had higher intake of snacks (crude OR = 1.403, 95% CI: 1.071~1.836, *p* < 0.05) and SSBs (crude OR = 1.540, 95% CI: 1.206~1.967, *p* < 0.001) than children with screen time < 1 h. These associations changed after the adjustment for sociodemographic confounders, children’s eating behaviors and parental feeding practices and the results are shown in Table 3. In model 1, after adjusting the sociodemographic factors and children’s eating behaviors (the associations between screen time and children’s eating behaviors are shown in Appendix A), the association between vegetables (AOR1 = 1.318, 95% CI: 1.061~1.637, *p* < 0.05), snacks (AOR1 = 1.406, 95% CI: 1.072~1.844, *p* < 0.05) and SSBs (AOR1 = 1.540, 95% CI: 1.206~1.967, *p* < 0.01) remained significant, while the association between screen time and fruits disappeared (AOR1 = 1.210, 95% CI: 0.986~1.486, *p* > 0.05). With the adjustment of parental feeding practice (the associations between screen time and parental feeding practice are shown in Appendix A) in model 2, the association between screen time and vegetables (AOR2 = 1.259, 95% CI: 1.011~1.568, *p* < 0.05) and SSBs (AOR2 = 1.411, 95% CI: 1.098~1.814, *p* < 0.01) remained statistically significant. The more screen time children had, the more SSB consumption and the less vegetable intake.

Vegetables ≥ once/day was used as a reference. Model 1 was adjusted according to the primary caregivers’ education, residence, children eating behavior (enjoyment of food, satiety responsiveness, food fussiness, desire to drink) and food neophobia. Model 2 was adjusted according to model 1 plus feeding practice (monitoring).

Intake of fruits ≥ once/day was used as a reference. Model 1 was adjusted according to the residence, children’s eating behavior (enjoyment of food, satiety responsiveness, slowness in eating, food fussiness) and food neophobia. Model 2 was adjusted according to model 1 plus feeding practice (monitoring).

Intake of snacks < once/week was used as a reference. Model 1was adjusted according to the children’s eating behavior (enjoyment of food, satiety responsiveness, emotional under-eating). Model 2 was adjusted according to model 1 plus feeding practice (restriction, food as reward and monitoring).

Intake of SSBs < once/week was used as a reference. Model 1 was adjusted according to the caregivers, children’s eating behavior (food responsiveness, emotional-over eating, desire to drink, food fussiness). Model 2 was adjusted according to model 1 plus feeding practice (perceived responsibility, concern, restriction, pressure to eat, monitoring).

## 4. Discussion

This study attempted to explore the association between screen time and food consumption among preschool children and found that the longer the screen time of preschoolers, the less the consumption of vegetables and fruits and the higher the consumption of snacks and SSBs.

### 4.1. The Status of Screen Time among Preschool Children

This study found that the average time of preschoolers using electronic screens was 1.36 ± 1.26 h, and 54.3% of preschoolers used electronic screens for more than 1 h. About half of the preschool children’s had screen time higher than the recommended standard (≥1 h per day), which is similar to the reports in some domestic cities, such as Guangzhou [17], Wuxi [38] and Shandong Province [39], etc., but still higher than Xinjiang [40] and lower than Shanghai [41], Beijing [42], Hong Kong [43] and other economically developed regions. When compared with Western countries [44,45,46], the proportion of preschooler’s screen time (≥1 h) was low in our study. A survey in the UK [47] reported that 79.4% of preschoolers aged 5 used electronic screens for more than 1 h, which was much higher than that of the children in this research. Preschoolers in different regions of the world have different amounts of screen time, which may be mainly due to the difference in economic levels [48], indicating that the issue of children’s screen time should be paid special attention in economically developed regions. It is worth noting that COVID-19 in the study site (Changsha City, Hunan Province) was well controlled; there was not any lockdown during the investigation time (from January to March, 2021). Thus, the pandemic had little impact on preschoolers’ screen time in this study. Additionally, screen time may differ significantly depending on the age range. However, the majority of the existing recommendations lack more specific requirements and are instead for a wider range of age groups, e.g., preschoolers [27]. The early years are a crucial time for visual development. Future research may look into the proper screen time guidelines for children in more specific age groups.

### 4.2. Screen Time and Its Associated Factors

The current study found a negative relationship between education level of primary caregivers and preschoolers’ screen time, which was similar to previous studies [49]. Well-educated primary caregivers can gain more knowledge about parenting and the dangers of screen time exposure; hence, they would set limits on screen time, resulting in better screen time management for preschoolers [50]. In addition, primary caregivers with higher educational levels are mostly employed and they spend less time on screens than that of stay-at-home parents, which can potentially affect children’s screen time [51]. Moreover, the higher the education level of the caregiver, the more attention they would pay to the influence of parents’ behavior on children, and consciously reduce the screen time to create a positive and healthy parenting environment [44,46,47]. In addition to the education level of the primary caregivers, other parents’ factors [43,48,49] can also influence children’s screen time, such as TV rules [52], which had a negative association with children’s screen time. Thus, we can focus on parental factors and develop some intervention methods to reduce children’s screen time.

### 4.3. The Status of F&V Consumption and Its Associated Factors

In our study, the consumption of vegetables (≥1 time(s)/day) was relatively higher than that of fruits (≥1 time(s)/day). The vegetable consumption was similar to previous studies, while the fruit consumption was lower than previous studies. In a domestic study conducted in Hangzhou, China, it was reported that 66.4% and 72.8% of preschool children consumed vegetables and fruits every day, respectively [53]. However, according to a survey in a Brazilian population, approximately 60% and 45% of children consumed vegetables and fruits (five or more times a week), respectively [54]. Preschoolers in many countries do not consume enough fruits and vegetables, such as in America and European countries [55,56]. The characteristics of vegetables and fruits themselves, such as odor and taste, may potentially affect preschool children’s liking for them [28]. Children’s eating behaviors, such as food fussiness, can also have an impact on preschoolers’ F&V consumption, which was confirmed in this study. Vegetables and fruits are the most common food that are rejected by children with food fussiness [57,58], as a result of a predisposition to taste [59]. Hence, all these factors (characteristics of F&V, such as odor, smell etc. and children’s eating behaviors (such as food fussiness) contribute to the low consumption of fruits and vegetables among preschoolers worldwide.

### 4.4. Association between Screen Time and Consumption Frequency of F&V

Except for these factors, studies have also shown that screen time has a negative impact on vegetable and fruit consumption [60,61,62] and the reason may be due to the fact that children may unconsciously consume more snacks and SSBs while using electronic screens, which in turn leads to low consumption of fruits and vegetables at mealtimes, especially for vegetables, due to the availability of unhealthy competitive food options [63,64]. Moreover, screen time was negatively correlated with physical activity time [62]. Children who spent more time with electronic screens were found to have less physical activity, less physical energy expenditure and less hunger, which also lead to a reduction in vegetable and fruit consumption. Thus, it is crucial for us to focus on the impact of screen time on food consumption among preschool children and consider the traits of vegetables, fruits and children’s eating behaviors when implementing food consumption interventions for preschoolers. Considering these factors, school gardening activities may be an excellent method to develop and improve preschoolers’ vegetable and fruit consumption [65,66].

### 4.5. The Status of Snacks and SSB Consumption and Its Associated Factors

In our study, the consumption of snacks and SSBs was 1.21 ± 2.58 times/week and 4.66 ± 6.40 times/week, respectively. Compared to a study among toddlers in China [28], the snack (3.25 ± 3.54 times/week) and SSB (0.45 ± 1.14 times/week) consumption levels were relatively higher, which maybe mainly due to increasing age [67]. However, compared with children of the same age (3~6 years old), the consumption of snacks and SSBs was lower than in the previous study, especially for SSBs [68,69,70,71]. Different racial/ethnic groups, availability of snacks, SSBs and feeding practice of parents may lead to these differences [72].

### 4.6. Association between Screen Time and Consumption Frequency of Snacks and SSBs

Screen time was positively associated with snack and SSB consumption, as confirmed in our study and is generally consistent with previous studies [6,7,58,60]. On one hand, preschool children may unconsciously consume foods with high energy density and low nutrient content, such as snacks and SSBs when using electronic screens [73]. On the other hand, advertisements on electronic screens may also influence children’s snack and SSB intake [74]. According to a previous study, food advertisements were the most common category on children’s TV, and the vast majority of them were SSBs or snacks ads [75], which could potentially influence children’s food choice. In addition, previous studies have confirmed a negative correlation between screen time and sleep duration in children [76]. A study [77] showed that children that slept for shorter periods of time consumed more calories at night. Children with more screen time are more likely to feel hungry due to shorter sleep durations and may turn to snacks or SSBs to fill their stomachs [78]. 

In addition, our study showed that compared with children’s eating behaviors, parental feeding practice had a greater impact on preschoolers’ food consumption, especially for snack consumption. On one hand, correct, active parental feeding practices could have positive effects on children’s food consumption and promote children to consume more healthy food and develop better diet quality [79,80,81]. On the other hand, inappropriate parental feeding practice can lead children to consume more salty and sugary snacks, resulting in poorer diet quality [82,83,84]. Meanwhile, parents may be overlooking the effects of SSBs on their children’s health. Studies have also shown that excessive SSB consumption will lead to dental caries [85], weight gain, risk of type 2 diabetes (T2D), cardiovascular disease (CVD), and some cancers [86]. Thus, researchers should not only focus on the influence of parental feeding practices on children’s food intake, but also pay attention to the problem of excessive intake of SSBs.

### 4.7. Strengths and Limitations

Our research had several strengths. This study employed CFQ, CEBQ, and CFNS, which are widely used and well-modified standard tools to assess parental feeding practices, children’s eating behaviors and children food neophobia, so the accuracy of our data was guaranteed. A relatively large sample also improved the representativeness of our study. In addition, the investigation tool was reliable. Participants were required to finish the questionnaire before submitting it, which ensured the integrity of the data. Nonetheless, our study had several limitations. This was a cross-sectional study, which was unable to demonstrate the cause–effect relationship. Furthermore, we used qualitative FFQ to collect dietary data, but unfortunately, it could not collect precise data on the food intake and calorie estimates; therefore, food intake should be paid more attention in future studies. Additionally, we also did not collect information about the frequency and timing of physical activity. Future research should investigate and assess levels of physical activity in preschoolers. Furthermore, the screen time of preschoolers was reported by their parents, who may not be aware of their children’s screen time in kindergarten. There may also be some items that parents cannot fully understand, resulting in errors.

## 5. Conclusions

Screen time exposure needs to be monitored in preschool children. We confirmed that preschoolers’ screen time was negatively associated with their consumption of vegetables and fruits and positively with snacks and SSBs. In addition, parental feeding practices and children’s eating behaviors also play indispensable roles. Future research should focus more on the impact of screen time on children’s unhealthy behaviors and dietary patterns. Furthermore, effective approaches that target screen time and parental feeding practices should be developed.

## Figures and Tables

**Figure 1 nutrients-14-04086-f001:**
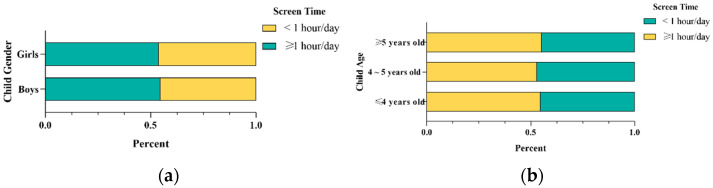
The age and gender distribution of screen time among preschool-age children. (**a**) Description of screen time ration by preschoolers’ gender. (**b**) Description of screen time ratio by preschoolers’ age.

**Figure 2 nutrients-14-04086-f002:**
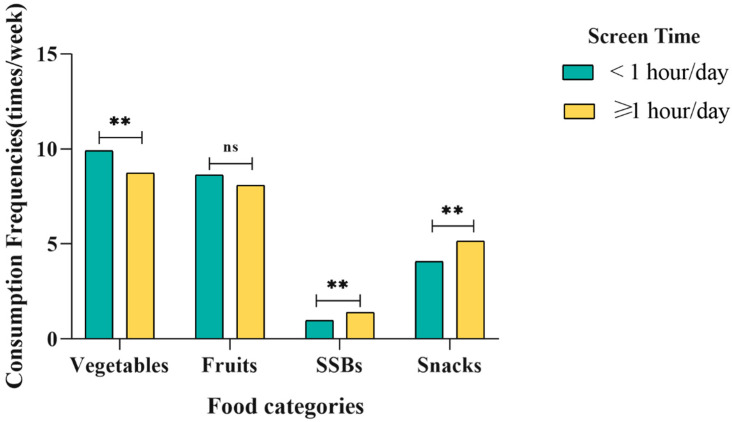
The distribution of consumption frequencies of vegetables, fruits, SSBs and snacks by screen time among preschool children. (times/week). ** *p* < 0.01, ns means no statistically significant, *p* ≥ 0.05.

**Table 1 nutrients-14-04086-t001:** Daily screen time by children’s characteristics (*n* = 1567, %).

Variables	Total 1567 (100)	Screen Time		*p*
		<1 h/Day	≥1 h/Day	
		716 (45.7)	851 (54.3)	
Gender				0.800
Boys	836 (53.4)	379 (52.9)	457 (53.7)	
Girls	731 (46.6)	337 (47.1)	394 (46.3)	
Age				0.776
≤4 years old	843 (53.8)	382 (53.4)	461 (54.2)	
4~5 years old	431 (27.5)	203 (28.4)	228 (26.8)	
≥5 years old	293 (18.7)	131 (18.3)	162 (19.0)	
Only child				0.672
Yes	605 (38.6)	281 (39.2)	324 (38.1)	
No	962 (61.4)	435 (60.8)	527 (61.9)	
Nationality				0.227
Han	1447 (92.3)	668 (93.3)	779 (91.5)	
Other	120 (7.7)	48 (6.7)	72 (8.5)	
Residence				0.142
Urban	1472 (93.9)	680 (95.0)	792 (93.1)	
Rural	95 (6.1)	36 (5.0)	59 (6.9)	
Left children behind				0.782
None	1480 (94.4)	678 (94.7)	802 (94.2)	
Left children behind	87 (5.6)	38 (5.3)	49 (5.8)	
Weight status *				0.095
Underweight	213 (13.6)	96 (13.4)	117 (13.7)	
Normal weight	885 (56.5)	424 (59.2)	461 (54.2)	
Overweight	236 (15.1)	99 (13.8)	137 (16.1)	
Obesity	108 (6.9)	40 (5.6)	68 (8.0)	
Caregiver				0.473
Mother	1392 (88.8)	641 (89.5)	751 (88.2)	
Father	175 (11.2)	75 (10.5)	100 (11.8)	
Caregiver’s nationality				0.104
Han	1487 (94.9)	687 (95.9)	800 (94.0)	
Other	80 (5.1)	29 (4.1)	51 (6.0)	
Caregiver’s education				**<0.001**
Junior high school or below	50 (3.2)	19 (2.7)	31 (3.6)	
High school	170 (10.8)	51 (7.1)	119 (14.0)	
College or above	1347 (86.0)	646 (90.2)	701 (82.4)	

* indicates 125 missing data. Electronic screen time was classified according to Chinese preschool children (3~6 years old) exercise guidelines. Compared by chi-square test. Bold values indicate statistically significant values, *p* < 0.05.

**Table 2 nutrients-14-04086-t002:** Daily food consumption by children’s age and gender among preschoolers (*n* = 1567, *n* (%)).

Food Categories	Total 1567 (100)	Child Age			*p*	Child Gender		*p*
		≤4	4~5	≥5		Boys	Girls	
		843 (53.8)	431 (27.5)	293 (18.7)		836 (53.4)	731 (46.6)	
Vegetables					0.124			0.816
≥1 time (s)/day	1009 (64.4)	548 (65.0)	262 (60.8)	199 (67.9)		541 (64.7)	468 (64.0)	
<1 time (s)/day	558 (35.6)	295 (35.0)	169 (39.2)	94 (32.1)		295 (35.3)	263 (36.0)	
Fruits					0.742			1.000
≥1 time (s)/day	902 (57.6)	479 (56.8)	249 (57.8)	174 (59.4)		481 (57.5)	421 (57.6)	
<1 time (s)/day	665 (42.4)	364 (43.2)	182 (42.2)	119 (40.6)		355 (42.5)	310 (42.4)	
Snacks					0.719			0.217
≤once/week	254 (16.2)	142 (16.8)	65 (15.1)	47 (16.0)		145 (17.3)	109 (14.9)	
>once/week	1313 (83.8)	701 (83.2)	366 (84.9)	246 (84.0)		691 (82.7)	622 (85.1)	
SSBs					0.235			0.964
≤once/week	1205 (76.9)	662 (78.5)	321 (74.5)	222 (75.8)		642 (76.8)	563 (77.0)	
>once/week	362 (23.1)	181 (21.5)	110 (25.5)	71 (24.2)		194 (23.2)	168 (23.0)	

Compared using chi-square test.

**Table 3 nutrients-14-04086-t003:** Binary logistic models of the association between screen time and consumption of FV, snacks and SSBS among preschool children (*n* = 1567).

Variables	Vegetables (OR, 95% CI)	Fruits (OR, 95% CI)	Snacks (OR, 95% CI)	SSBs (OR, 95% CI)
	Crude OR			
Screen time (<1 h/day = *Ref*)				
≥1 h/day	**1.371 (1.112, 1.690) ****	**1.233 (1.008, 1.509) ***	**1.403 (1.071, 1.836) ***	**1.566 (1.230, 1.993) *****
	Model 1			
Screen time (<1 h = *Ref*)				
≥1 h/day	**1.318 (1.061, 1.637) ***	1.210 (0.986, 1.486)	**1.406 (1.072, 1.844) ***	**1.540(1.206, 1.967) ****
	Model 2			
Screen time (<1 h/day = *Ref*)				
≥1 h/day	**1.259 (1.011, 1.568) ***	1.156 (0.939, 1.423)	1.279 (0.969, 1.687)	**1.411 (1.098, 1.814) ****

Bold values indicate statistically significant, * *p* < 0.05, ** *p* < 0.01, *** *p* < 0.001. *R**ef* means as reference.

## Data Availability

The data that support the findings of this study are not publicly available, due to the data containing information that could compromise the participants’ privacy, but are available from the corresponding author upon reasonable request.

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
