# Peer review of "Screen Time and Its Association with Vegetables, Fruits, Snacks and Sugary Sweetened Beverages Intake among Chinese Preschool Children in Changsha, Hunan Province: A Cross-Sectional Study"

_nutrients, 2022, doi:10.3390/nu14194086_

Round 1

Reviewer 1 Report

The study is well written and drawn. I suggest that the English language be revised.

Although the study does not represent elements of strong originality, I believe that it has the criteria to be accepted and published after revision. Because it is representative of habits currently of interest to the community

Reviewer 2 Report

The manuscript submitted for publication to Nutrients by Huo et al., titled: "Screen time and its association with vegetables, fruits, snacks and sugary sweetened beverages intake among Chinese preschool children in Changsha, Hunan Province: A Cross-Sectional study" is an interesting observational study, aiming to investigate the relationship between screen time and the quality of diet in the concept of fruits and vegetables versus snacks and sugary beverages, in specific region in China.

This is an interesting topic and the reviewer would like to offer the following points to be considered by the authors:

1. The authors appear to utilize questionnaires. Were these questionnaires validated for the population applied to? 

2. The authors enrolled children 2-7 years old. However the recommendations from a dietary perspective are quite different in the 2 versus 7 years old. Also, screen practices may vary significantly in this age range. 

3. Is there any assessment in terms of physical activity of children enrolled in the study?

4. The physical activity levels of 2 year olds may be significantly different than that of 7 years old. How did the authors control for that?

5. What was the caloric intake for the children enrolled? How are these different in the different groups?

6. English language, syntax, grammar and flow need improvement. The reviewer suggests that an English native speaker works on the manuscript.

Reviewer 3 Report

General

The manuscript by Huo et al. covers an important topic of screen time and its association with fruits, vegetables and sugary sweetened beverages in children which appropriate eating habits are a matter of great concern because they shape further dietary habits during adolescence and adulthood. The article is generally well-written, however, I have some objections which are listed below. Please pay also attention to English language because there are some errors which must be corrected. I would suggest to have your manuscript proofread by a professional native speaker.

Abstract

-Line 23 – “included” instead of “include”

-Line 28 – “children’s” instead of “children”

Introduction

-Line 37 – “improvement” instead of “improve”

-Line 43 – Authors have to clarify what is the standard/proper fruits and vegetables intake (perhaps according to Chinese recommendations).

-Line 44 – I would suggest to rephrase the sentence, for example: “Other studies have also shown these problems, namely the deficiency of F&V’s […]”.

-Line 47 – maybe you should use the word “negatively” instead of “directly”?

-Line 52 – Authors write that “Studies show”, however, there didn’t provide any references here.

-Lines 55-57 – Are these negative consequences related to adults or children? You have to clarify it.

-Line 59 – Please add information whether this is Chinese studies or from another country.

-Lines 59-60 – rephrase the sentence because it is not grammatically correct.

-Line 64 – “decreased”, not “decreasing”

Materials and Methods

-Line 89 – while describing applied methods, please be consistent and use Past Simple.

-Line 91 – “were”, not “are”.

-Authors write that the study was conducted in the period from January to March 2021. It is crucial, whether on that time, there were any restrictions implemented in Hunan Province due to the COVID-19 pandemic. If so, Authors have to meticulously describe them because they may greatly influence the obtained results. For example, if there was a lockdown during January-March 2021 and children were staying at home, it is very likely that children spent more time watching TV or playing computer games.

-Did Authors perform calculation of sample size?

-There is little information regarding applied questionnaires. Were these questionnaires validated in the group of preschoolers?

-Line 133 – what was the purpose of distinguishing in the FFQ fruits and vegetables rich in vitamin A and the others?

-Lines 134-135 – the reference is missing.

-Line 138 – maybe “included”, instead of “contains”?

-Lines 139-141 – reference is missing.

-Lines 202-204 – please rephrase this sentence because it is grammatically incorrect.

Discussion

-While discussing the results in terms of the Status of Screen Time among Preschool Children you have to take into consideration the period in which your study was conducted. As I mentioned, strict lockdown in 2021 may have influenced the registered screen time.

Round 2

Reviewer 2 Report

The authors have made a reasonable effort in addressing reviewer's points. Proofreading is suggested.

Reviewer 3 Report

Dear Authors,

Thank you a lot for incorporating my suggestions into the manuscript.

I would suggest to add in the manuscript the purpose of distinguishing in the FFQ fruits and vegetables rich in vitamin A and the others - just like you did it in the response to me. It will be more clear for the readers.

Author Response

Response to Reviewer 3 Comments

Dear Reviewer,

Thanks very much for your careful reading and valuable comments. You help us a lot in revising and improving our paper. On behalf of my co-authors, we would like to express our great appreciation to you.

Here we submit a new version of our manuscript, revised according to the suggestions, with the title “Screen time and its association with vegetables, fruits, snacks and sugary sweetened beverages intake among Chinese pre-school children in Changsha, Hunan Province: A Cross-Sectional study” (Manuscript ID: Nutrients-1929698). You will see the difference made to the revised manuscript.

If you have any questions about this paper, please do not hesitate to contact me.

Corresponding author: Qian Lin Ph.D. E-mail: linqian@csu.edu.cn

Sincerely yours,

                                                                    Qian Lin

Here are point-to-point answers to your detailed comments.

Point 1: I would suggest to add in the manuscript the purpose of distinguishing in the FFQ fruits and vegetables rich in vitamin A and the others - just like you did it in the response to me. It will be more clear for the readers.

Response 1: Thank you so much for your suggestions. We have added two sentences in the section of method, to clarify the purpose of distinguishing in the FFQ fruits and vegetables rich in vitamin A and the others as follows:

“Considering the prevalence of marginal vitamin A deficiency was 31.53% in Chinese preschool children [30], we distinguish vegetables and fruits into vitamin A-rich vegetables/other vegetables and vitamin A-rich fruits/other fruits to know the current status of vitamin A-rich food intake among preschool children.” please check line 141-144.

[30]         Song, P.; Wang, J.; Wei, W.; Chang, X.; Wang, M.; An, L. The Prevalence of Vitamin A Deficiency in Chinese Children: A Systematic Review and Bayesian Meta-Analysis. Nutrients 2017, 9, doi:10.3390/nu9121285.
